Stability of hepatitis B virus pregenomic RNA in plasma specimens under various temperatures and storage conditions

Rattanachaisit Pakkapon 1 2
Suksawatamnuay Sirinporn 3 4
Sriphoosanaphan Supachaya 1 3
Thanapirom Kessarin 1 4
Thaimai Panarat 3 4
Siripon Nipaporn 3 4
Sittisomwong Sukanya 3 4
Poovorawan Yong 5
Komolmit Piyawat pkomolmit@yahoo.co.uk 1 3 4
1 Division of Gastroenterology, Department of Medicine, Faculty of Medicine, Chulalongkorn University and King Chulalongkorn Memorial Hospital , Bangkok , Thailand
2 Department of Physiology, Faculty of Medicine, Chulalongkorn University , Bangkok , Thailand
3 Center of Excellence in Liver Diseases, King Chulalongkorn Memorial Hospital, Thai Red Cross Society , Bangkok , Thailand
4 Liver Fibrosis and Cirrhosis Research Unit, Chulalongkorn University , Bangkok , Thailand
5 Center of Excellence in Clinical Virology, Faculty of Medicine, Chulalongkorn University , Bangkok , Thailand
Böttcher Bettina
Electronic publication date: 2021 Apr 14
Publication date: 2021
Volume: 9
Electronic Location ID: e11207
Received 2020 Oct 15; Accepted 2021 Mar 12
Copyright: ©2021 Rattanachaisit et al.
Copyright year: 2021
Copyright holder: Rattanachaisit et al.
License: This is an open access article distributed under the terms of the Creative Commons Attribution License, which permits unrestricted use, distribution, reproduction and adaptation in any medium and for any purpose provided that it is properly attributed. For attribution, the original author(s), title, publication source (PeerJ) and either DOI or URL of the article must be cited.
License URL: https://creativecommons.org/licenses/by/4.0/

Keywords: Pregenomic RNA, Hepatitis B, Stability

Funding: Ratchadaphiseksomphot Endowment Fund of hepatic fibrosis and cirrhosis research unit GRU 6105530009-1 The Research Chair Grant from the National Science and Technology Development Agency P-15-50004 Center of Excellence in Clinical Virology at Chulalongkorn University and the MK Restaurant Group Public Company Limited This work was funded by the Ratchadaphiseksomphot Endowment Fund of hepatic fibrosis and cirrhosis research unit (GRU 6105530009-1), the Research Chair Grant from the National Science and Technology Development Agency (P-15-50004), the Center of Excellence in Clinical Virology at Chulalongkorn University and the MK Restaurant Group Public Company Limited. The funders had no role in study design, data collection and analysis, decision to publish, or preparation of the manuscript.

==============================
Background

Hepatitis B virus (HBV) pregenomic RNA (pgRNA) has gained increasing attention owing to its role in replication of covalently closed circular DNA (cccDNA) in HBV. This marker has the potential to be used in clinical programs aimed to manage HBV infections. However, several reports on HBV pgRNA levels in clinical cases have conflicting results. RNA is easily degraded when exposed to heat and other environmental stressors. However, the stability of HBV pgRNA, during blood sample collection before the standard automated quantification, has never been estimated. This study aimed to demonstrate the effect of two different temperature conditions and storage durations on the stability of HBV pgRNA.

Method

Blood from forty patients with chronic hepatitis B infection, who also showed evidence of active HBV DNA replication, was collected and processed within 2 h of collection. Plasma from each patient was divided and stored at 4 °C and 25 °C (room temperature) for six different storage durations (0, 2, 6, 12, 24, and 48 h) and subsequently transferred to −80 °C for storage. The effect of multiple cycles of freezing and thawing of plasma at −20 °C or −80 °C was evaluated using samples from ten patients. Quantification of pgRNA from the samples was performed simultaneously, using the digital polymerase chain reaction (dPCR) method. The differences in pgRNA levels at baseline and each time point were compared using generalized estimating equation (GEE). A change greater than 0.5 log10 copies/mL of pgRNA is considered clinically significant. Statistical analyses were conducted using Stata 16.0.

Results

The mean HBV pgRNA level in the initially collected plasma samples was 5.58 log10copies/mL (ranging from 3.08 to 8.04 log10 copies/mL). The mean pgRNA levels in samples stored for different time periods compared with the initial reference sample (time 0) significantly decreased. The levels of pgRNA for 6, 12, 24, and 48 h of storage reduced by −0.05 log10 copies/mL (95% confidence interval (CI) −0.095 to −0.005, p = 0.03), −0.075 log10 copies/mL (95% CI [−0.12 to −0.03], p = 0.001), −0.084 log10 copies/mL (95% CI [−0.13 to −0.039], p =  < 0.001), and −0.120 log10 copies/mL (95% CI [−0.17 to −0.076], p =  < 0.001), respectively. However, these changes were below 0.5 log10 copies/mL and thus were not clinically significant. Compared with the samples stored at 4 °C, there were no significant differences in pgRNA levels in samples stored at 25 °C for any of the storage durations (−0.01 log10 copies/mL; 95% CI [−0.708 to 0.689], p = 0.98). No significant difference in the levels of pgRNA was observed in the plasma samples, following four freeze-thaw cycles at −20 °C and −80 °C.

Conclusion

The plasma HBV pgRNA level was stable at 4 °C and at room temperature for at least 48 h and under multiple freeze-thaw cycles. Our results suggest that pgRNA is stable during the process of blood collection, and therefore results of pgRNA quantification are reliable.

Introduction

Chronic hepatitis B virus (HBV) infection is one of the leading causes of morbidity and mortality worldwide (WHO, 2017). Serum levels of HBV DNA and HBV surface antigen (HBsAg) are used to evaluate the efficacy of HBV treatment (Terrault et al., 2018). However, there are some limitations of their use in indicating the extent of HBV DNA replication. Recently, HBV pregenomic RNA (pgRNA) quantification is being used as a diagnostic marker to represent HBV cccDNA replication (Liu et al., 2019).

The HBV genome is a 3.2 kb long, relaxed, circular, and partially double-stranded DNA (Datta et al., 2012). After attaching to and entering the hepatocytes through endocytosis, the viral particles release the relaxed circular DNA (rcDNA) into the nucleus. The rcDNA is repaired to form a covalently closed circular DNA (cccDNA), which becomes the template for transcription of viral RNAs (Ganem & Schneider, 2001; Seeger & Mason, 2000). Viral RNAs transcribed from cccDNA consist of pregenomic RNA and subgenomic RNA. Pregenomic RNA is a template for reverse transcription of viral DNA and itself translates into precore, core, and polymerase proteins (Datta et al., 2012). New viral particles are assembled from pregenomic RNA, using viral polymerases, in the cytoplasm. Viral polymerase reverse-transcribes some of the pgRNA back to relaxed circular DNA. This viral particle is either released from hepatocytes through the endoplasmic reticulum and Golgi complex modification or travels back to the nucleus for amplification of the cccDNA pool (Datta et al., 2012; Rehermann & Nascimbeni, 2005). A recent study showed that pgRNA is also encapsidated and released from infected hepatocytes (HBV RNA virion-like particles) (Jansen et al., 2016; Wang et al., 2016).

Circulating serum HBV RNAs are heterogenous, with the major component being pgRNA species, which are localized either in the unenveloped capsid or virion (Bai et al., 2018). The encapsidated forms are detergent- and ribonuclease-resistant (Shen et al., 2020) and they bind with specific antibodies, forming capsid-antibody complexes (Bai et al., 2018). HBV pgRNA variants are of varying lengths, with some undergoing post-transcriptional splicing at the 5′ terminal, while the others are 3′ terminally truncated forms (Shen et al., 2020). There is no evidence for the infectivity of these particles (Lu et al., 2017). HBV RNA-virions do not induce a productive infection in vitro, unlike the HBV DNA-virions (Shen et al., 2020).

Pregenomic RNA level is used as an indicator of HBV cccDNA replication. Theoretically, this novel marker would be superior to HBsAg (HBsAg (q)), since the latter can be synthesized from both cccDNA and integrated DNA, resulting in decreased specificity as marker for viral replication (Cornberg et al., 2017). Clinical applications of quantitation of pgRNA levels include monitoring of the treatment efficacy of nucleot(s)ide analogs (NAs) and interferons and determination of when to terminate NA treatment (Huang et al., 2015; Jansen et al., 2016; Tsuge et al., 2013).

In general, RNA is easily degraded by various factors during quantitation, including temperature, storage time, ribonuclease (RNase), or contamination with inhibitors (Sanders et al., 2018). RNA stability was confirmed before the clinical application of various RNA assays, such as those for human immunodeficiency virus (HIV) and hepatitis C virus (HCV). HIV RNA in plasma can be stored at room temperature (25 °C) and 4 °C for up to 3 days (Sebire et al., 1998), and HCV RNA in blood remains stable at 4 °C for up to 4 days but stability decreases during storage at 23 °C or 37 °C (De Moreaude Gerbehaye et al., 2002; Krajden et al., 1999).

HBV pgRNA stability under different temperatures and storage durations has not been previously estimated. In addition, stability of the encapsidated form of HBV RNA virion-like particles in blood samples during laboratory processes remains unknown. These estimates are crucial for the use of HBV pgRNA in comparative clinical studies and diagnosis. Thus, we aimed to instigate the effect of two different temperature conditions and storage durations on the stability of HBV pgRNA.

Material and Methods

This study was approved by the Institutional Review Board of Faculty of Medicine, Chulalongkorn University (IRB number 765/61).

Patient characteristics

Adult subjects aged 18 years or older, who had chronic hepatitis B infection were enrolled from The Liver Clinic, King Chulalongkorn Memorial Hospital, Bangkok, Thailand. Patients, selected for the study, showed evidence of active HBV DNA replication, defined by plasma HBV DNA of more than 2,000 IU/mL or quantitative HBsAg of more than 2,000 IU/mL. The exclusion criteria included, (1) coinfection with HIV or HCV and (2) unwillingness to participate in this study. Forty patients met the criteria and were included in this study. All participating patients provided informed written consent.

Sample collection and processing

Twenty milliliters of blood from the participating patients was collected in EDTA tubes. The plasma was separated at 4 °C by centrifugation within 2 h of collection. Plasma samples from each patient were divided into 11 aliquots and stored at 4 °C and 25 °C (room temperature) for six different storage durations (0, 2, 6, 12, 24, and 48 h) and subsequently transferred to −80 °C for storage.

Effect of freezing and thawing of plasma

The effects of multiple freeze-thaw cycles on HBV pgRNA at −20 °C and −80 °C were assessed. Aliquots of ten plasma samples were stored at −20 °C and −80 °C. A freeze-thaw cycle was defined as freezing at −20 °C or −80 °C for 22 h and thawing at room temperature for 2 h. For each sample set, at each temperature, four freeze-thaw cycles were performed. One aliquot was used for quantification after every freeze-thaw cycle.

RNA extraction and quantification

Total RNA from plasma (200 µL) was extracted using QIAsymphony DSP Virus/Pathogen Mini Kit (Cat No. 937036; Qiagen Gmbh, Germany) and treated with DNAseI (Cat No. 79256; Qiagen Gmbh) on the QIAsymphony SP/AS instruments, following the manufacturer’s instructions. Isolated HBV RNA was reverse transcribed using ImProm-II Reverse Transcriptase (Cat No. A3802; Promega, Madison, WI, USA), with HBV-specific RT primers (5-ATTCTCAGACCGTAGCACACGACACCGAGATTGAGATCTTCTGCGAC-3) as previously described (Wang et al., 2016). To ensure that HBV DNA does not interfere with the RNA estimation, a control PCR, using the RNA samples without reverse transcription as template, was performed. HBV pgRNA was quantified using the Droplet Digital PCR System (ddPCR) (Wang et al., 2018) (QX200; Bio-Rad Laboratories, Inc., California). The PCR mixture (20 µL) consisted of 10 µL of 2x supermix for probes (Cat No. 1863010; Bio-Rad Laboratories, Inc., Hercules, CA, USA), 1.8 µL of each primer (10 nM), 1 µL of TaqMan probe (5 nM), 2 µL of the cDNA template, and 3.4 µL of water. The primers and probe used were, forward primer: 5-AYAGACCATCAAATGCCC-3; reverse primer: 5-ATTCTCAGACCGTAGCACACGACAC-3 and probe: 5-FAM-CTTATCAACACTTCCGGARACTACTGTTGTTAGAC-BHQ1-3 (Wang et al., 2016). PCR mixture and droplet generation oil (70 µL) were added to the DG32 cartridge. The droplets were produced through the automated droplet generator of the QX200 Droplet Digital PCR system (Bio-Rad). Each droplet was transferred to a 96-well PCR plate for amplification using the Mastercycler® Pro Thermal Cyclers (Eppendorf). The following protocol was used: one cycle at 95 °C for 10 min, 40 cycles at 94 °C for 30 s and 56 °C for 1 min, and one cycle at 98 °C for 10 min. The concentration of HBV pgRNA was estimated using QuantaSoft™ version 1.7 (Bio-Rad Laboratories, Inc.). The lower limit of detection was determined as 100 copies/mL. The mean lowest quantity (copies/mL) was estimated through serial dilution of specimens until the levels became undetectable in most of the diluted aliquots.

Statistical analysis

Statistical analysis was conducted using Stata 16.0. Results were expressed as mean pgRNA concentration and mean change from baseline for each time duration at each temperature. Changes in pgRNA levels from baseline values were compared using a generalized estimating equation (GEE) adjusting for repeated measurements in each sample. To compare the stability at a particular temperature following freeze-thaw cycles, GEE was used. We calculated the change in ratio per freeze-thaw cycle, relative to 1 cycle at the corresponding temperature, which was used as the reference group. A p-value of 0.05 or less is considered statistically significant. However, a change in pgRNA level greater than 0.5 log copies/mL is considered clinically significant (Pawlotsky, 1997).

Results

From the initial forty plasma samples, HBV pgRNA levels ranged from 3.08 to 8.04 log10 copies/mL with a mean of 5.58 log10 copies/mL (Fig. 1). Changes in HBV pgRNA levels for different durations of storage and temperature are expressed, with mean, standard deviation (SD), minimum (min), median (P50), 25th percentile (P25), 75th percentile (P75), and maximum (max) values, in Table 1. Compared to the initial reference sample (time 0), using GEE, the mean changes in pgRNA levels were found to be statistically significant. The pgRNA levels of samples stored for 6, 12, 24, and 48 h decreased from the initial values and are presented in Fig. 2 and Table 2.

Since the differences in pgRNA levels for the different storage durations were within 0.5 log10 copies/mL, these changes are not clinically significant. There were also no significant changes in pgRNA levels between the two storage temperatures (−0.01 log10 copies/mL; 95% CI [−0.708 to 0.689], p = 0.98) (Fig. 2).

There was no significant difference in the levels of pgRNA in the samples, following the freeze-thaw cycles (Fig. 3). Four cycles of freezing and thawing did not result in significant ratio changes in HBV pgRNA levels in any of the samples tested ( p = 0.45 and 0.59 for −20 °C and −80 °C, respectively) (Table 3).

Discussion

HBV pgRNA levels are increasingly being used as a biomarker of intrahepatic cccDNA replication, and have been studied for their role in HBV pathogenesis and treatment. However, there exists no standard for collection, processing, and storage to ensure the accuracy and reproducibility of quantification tests (Liu et al., 2019).

Based on our results, we conclude that blood samples collected in EDTA, and processed at 4 °C within 2 h of collection can be stored at 4 °C and 25 °C for up to 48 h without significantly affecting pgRNA levels. In addition, there was no significant difference in the levels of HBV pgRNA following four freeze-thaw cycles. The aforementioned time duration for blood collection and processing can be applied to real-world situations, in which the time from blood collection to quantification is within 48 h before being shipped to a central laboratory for further processing. The two temperature points investigated in our study are frequently used for storage and transport of blood samples. In addition, multiple freeze-thaw cycles at −20 °C and −80 °C are frequently used in clinical research.

Figure 1 HBV pgRNA degradation over time and temperature.

The mean (+∕ − 1SE) of HBV pgRNA level in plasma stored at either 4 or 25 °C. HBV pgRNA loads, expressed in log10 copies/mL, were measured at 2, 6, 12, 24 and 48 h. Triangles indicate 4 °C storage and circles indicate 25 °C storage.

Table 1 Changes in HBV pgRNA for each storage duration and at two different temperatures, (A) 4 °C and (B) room temperature (25 °C).

Time (h)	N	Mean	SD	Min	P25	P50	P75	Max	p*	
(A)	
0	40	0	0	0	0	0	0	0	–	
2	40	−0.04	0.13	−0.45	−0.1	−0.02	0.05	0.16	0.29	
6	40	−0.02	0.15	−0.52	−0.09	0.00	0.06	0.35	0.54	
12	40	−0.1	0.21	−0.95	−0.17	−0.04	0.03	0.22	0.004	
24	40	−0.08	0.22	−0.83	−0.18	−0.03	0.05	0.37	0.03	
48	40	−0.11	0.26	−1.38	−0.2	−0.06	0.04	0.25	0.001	
(B)	
0	40	0	0	0	0	0	0	0	–	
2	40	−0.05	0.2	−0.71	−0.12	−0.01	0.08	0.32	0.13	
6	40	−0.08	0.17	−0.51	−0.13	−0.07	0.04	0.18	0.009	
12	40	−0.05	0.16	−0.37	−0.16	−0.04	0.07	0.39	0.09	
24	40	−0.09	0.22	−1.08	−0.13	−0.07	0.05	0.22	0.002	
48	40	−0.13	0.22	−0.95	−0.18	−0.08	0.00	0.16	<0.001	
Notes.

* Comparison of mean levels for each storage duration with initial levels (t = 0).

HBV pgRNA loads were expressed in log10 copies/mL.

h hours

SD standard deviation

min minimum

P25 25th percentile

P50 median

P75 75th percentile

max maximum

Figure 2 Changes in HBV pgRNA level from baseline.

The mean (+∕ − 1SE) of HBV pgRNA level change from baseline (time = 0) in plasma stored at either 4 or 25 °C. HBV pgRNA loads, expressed in log10 copies/mL, were measured at 2, 6, 12, 24 and 48 h. Circles indicate 4 °C storage and squares indicate 25 °C storage.

Table 2 Generalized estimating equation (GEE) results comparing the differences of mean baseline pgRNA level and mean pgRNA level for each storage duration.

Time (h)	Coefficient (95% CI)	p	
0	0 (Reference)	–	
2	−0.04 (−0.085 to 0.004)	0.08	
6	−0.05 (−0.095 to −0.005)	0.03	
12	−0.075 (−0.12 to −0.03)	0.001	
24	−0.084 (−0.13 to −0.039)	<0.001	
48	−0.120 (−0.17 to −0.076)	<0.001	
Notes.

h hours

CI confidence interval

For accuracy and consistency of the quantification process, the automated system for RNA extraction used in this study allowed for less human interference and RNase contamination. The co-extracted DNA that remained after DNase reaction was tested using HBV DNA-specific and housekeeping DNA primers. In the 8 selected samples, DNA did not remain after the DNase reaction.

However, there was a slight decrease in the HBV pgRNA level in samples 29 and 33 (Figs. S1 and S2, and Table S1). One explanation for this decrease might stem from an oxidation reaction (Relova et al., 2018). Thus, shorter period of handling before storage and controlled low-temperature transfer would reduce this variation.

Figure 3 HBV pgRNA concentration over multiple freeze-thaw cycles at −20 °C and −80 °C.

The mean (+/−1SE) of HBV pgRNA levels, expressed as log10 copies/mL, were measured for up to four freeze-thaw cycles. Triangles and circles indicate storage at −80 °C and −20 °C, respectively.

Some of the mean changes in HBV pgRNA levels, from the baseline levels, for storage duration and temperature were found to be significantly different (Table 1). A generalized estimating equation (GEE), adjusting for repeated measurements in each sample, was applied to build a statistical model for repeated measurements. The GEE findings indicated significant changes in pgRNA levels in samples with a storage duration beyond 6 h (Table 2). However, these findings were only an illustration of the statistical reliability. In terms of clinical application, these changes are acceptable and not significant, because clinical differences are considered significant only when the change in pgRNA levels is more than 0.5 log10 copies/mL (Baleriola et al., 2011; Pawlotsky, 1997).

Currently, there is no standard method for serum HBV pgRNA detection (Liu et al., 2019). Digital PCR (dPCR), in addition to its excellent sensitivity and quantification of the target molecule, does not need a standard controlled curve, as those required in real-time or other quantitative PCR assays (qPCR) (Cao et al., 2017). It also sensitive to very low levels of circulating nucleic acids in blood (Kuypers & Jerome, 2017). A study comparing HBV DNA detection using qPCR and dPCR showed lower copy number detection and better reproducibility of the dPCR assay, which improves sensitivity and specificity for serum HBV DNA measurements (Tang et al., 2016). In this study, we performed pgRNA quantification by dPCR to ensure that samples with very low titers of HBV pgRNA could be detected.

Table 3 Generalized estimating equation (GEE) results comparing the ratio changes per freeze-thaw cycle, relative to 1 cycle as the reference group.

Temperature and cycle number	Ratio change (95% CI)	P	
−80 °C		0.59	
1	1 (ref)		
2	1 (0.96–1.03)		
3	0.99 (0.96–1.03)		
4	1.02 (0.98–1.05)		
−20 °C		0.45	
1	1 (ref)		
2	1.01 (0.98–1.05)		
3	1.02 (0.99–1.06)		
4	1.03 (0.99–1.06)		

Species and forms of serum HBV RNA have not yet been clearly identified. Wang et al. reported that HBV RNA detected in the supernatant is pgRNA contained in the enveloped particles (Wang et al., 2016). However, more evidence is needed to confirm this (Liu et al., 2019). In addition, serum HBV RNA is heterogeneous, comprising an intact genome length of 3.5-kb along with spliced and polyA-free pgRNA, and varies depending on the stage of chronic HBV infection, medication administered, and detection methods (Hacker et al., 2004; Liu et al., 2019; Shen et al., 2020; Wang et al., 2016). We used an HBV-specific primer pair as described by Wang et al. (2016), which selectively amplified specific pgRNA species. The primers target a sequence upstream of DR2; the PCR amplicon is located between the 3′ splicing site and the RT primer site. This yields a more specific amplicon, with better coverage of the spliced and 3′ truncated forms, which are the predominant species of serum HBV pgRNA (Shen et al., 2020).

The stability of pgRNA observed in this study might reflect the nature of the HBV RNA itself, and indicates encapsidation of the internal RNA, thus improving its resistance to the environment.

This study had several limitations. In future investigations, only the robust reproduction of specimen handling, processing, and storage techniques, described in our study, can result in pgRNA stabilities similar to that reported in this study. Other storage times and temperatures cannot be applied. Second, we measured only pgRNA specific to the HBV pgRNA-specific primer. There are no data on variation in the levels of other HBV RNA species. Our method, however, can be applied as a future reference. Finally, this stability was observed in patients under the current approved HBV treatment. Patients recruited in this study were either NA naïve or under NA treatment at the time of the study. NA treatment does not interfere with the encapsidation of HBV virions. However, some novel anti-HBV drugs that target pgRNA, such as (Z)-2-(allylamino)-4-amino- N′-cyanothiazole-5-carboximidamide (AACC), inhibit pgRNA and polymerase action, resulting in capsid assembly inhibition and encapsidated pgRNA reduction (Jo et al., 2020). There is still a lack of data on HBV pgRNA levels in patients using these new drugs.

Conclusions

We have demonstrated the stability of the plasma level of HBV pgRNA at 4 °C and room temperature for up to 48 h and following multiple freeze-thaw cycles before it is quantified. Our results confirmed the stability of pgRNA using the established strategy of blood collection and storage, and thus, quantitation of pgRNA is reliable.

Supplemental Information

Figure S1 HBV pgRNA degradation over time under 4 °C storage

Each graph represents individual HBV pgRNA load of each patient (n = 40). HBV pgRNA loads, expressed in log10 copies/mL, were measured at 2, 6, 12, 24 and 48 h.

Click here for additional data file.

Figure S2 HBV pgRNA degradation over time under 25 °C storage

Each graph represents individual HBV pgRNA load of each patient (n = 40). HBV pgRNA loads, expressed in log10 copies/mL, were measured at 2, 6, 12, 24 and 48 h.

Click here for additional data file.

Supplemental Information 3 HBV pgRNA levels in all specimens

Click here for additional data file.

Supplemental Information 4 HBV pgRNA concentration over multiple freeze-thaw cycles at −20 °C and −80 °C

Click here for additional data file.

We would like to thank Professor Stephen Kerr, BPharm (Hons), MIPH, PhD, Director, Biostatistics Excellence Center, Faculty of Medicine, Chulalongkorn University, and Biostatistics at HIV-NAT for data analysis.

Additional Information and Declarations

Competing Interests

Author Contributions

Human Ethics

Data Availability

The authors declare there are no competing interests.

Pakkapon Rattanachaisit conceived and designed the experiments, performed the experiments, analyzed the data, prepared figures and/or tables, authored or reviewed drafts of the paper, and approved the final draft.

Sirinporn Suksawatamnuay, Panarat Thaimai, Nipaporn Siripon and Sukanya Sittisomwong performed the experiments, authored or reviewed drafts of the paper, and approved the final draft.

Supachaya Sriphoosanaphan, Kessarin Thanapirom, Yong Poovorawan and Piyawat Komolmit conceived and designed the experiments, authored or reviewed drafts of the paper, and approved the final draft.

The following information was supplied relating to ethical approvals (i.e., approving body and any reference numbers):

The Institutional Review Board of Faculty of Medicine, Chulalongkorn University approved this research (765/61).

The following information was supplied regarding data availability:

The raw measurements are available in the Supplemental Files.

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
