# Peer review of "Stability of hepatitis B virus pregenomic RNA in plasma specimens under various temperatures and storage conditions"

_PeerJ, doi:10.7717/peerj.11207_

## Round 0.1 · original submission · Minor Revisions

As you see, both reviewers are positive. Please revise your manuscript accordingly. In particular, provide more details on the RNA extraction and quantification as suggested by reviewer 1 and discuss the more significant differences presented in Table 2

·

Basic reporting

In the introduction section, the authors described the background information on serum HBV RNA. However, the molecular nature of serum HBV RNA is still under fierce debate. The authors are advised to update their knowledge with newer publications. Several references are recommended below:
PMID: 33079954, 30282709, 30362148

In my opinion, it is best not to restrict serum circulating HBV RNA as pgRNA as there are plenty of evidence against this assertion.

Experimental design

The overall design of this study is acceptable. However, the authors have explored very limited area in the stability of this biomarker.

In addition to two days of room temperature storage. There should also be investigation on the stability of serum HBV RNA after freeze-thaw cycles (-20℃ and -80℃) which is critical for the reliability of retrospective studies.

The authors should also put enough efforts in detailing the protocols for RNA extraction and quantification. The current form did not specify how the DNase I treatment was performed and inactivated, the exact product name and catalog number of Bio-Rad digital RT-PCR mix was not specified and how exactly the ddPCR was performed was largely unclear.

Validity of the findings

The findings and statistical analysis are valid.

Additional comments

The current manuscript should include more details on the ddPCR. The background of serum HBV RNA should also be updated. I advise that the authors should perform additional experiments on the stability of serum HBV RNA after more harsh conditions such as prolonged room temperature storage and freeze-thaw cycles in order to test the analytical limits of this novel biomarker.

Reviewer 2 ·

Basic reporting

The manuscript entitled, Stability of hepatitis B virus pregenomic RNA in plasma specimens under various temperatures and storage conditions” describes how the pgRNA level differs regarding different storage conditions. This subject is important as pgRNA reflects viral replication activity and is valuable for monitoring the effect in patients receiving novel anti-HBV therapies. Nevertheless, different storage conditions were supposed to affect RNA levels and in consequence, interfere with the quantification results. This study demonstrated that HBV pgRNA is quite stable under different storage temperatures.

Experimental design

no comment

Validity of the findings

These findings are important but the authors should discuss more significant differences presented in Table 2. The authors claim that pgRNA is stable at room temperature at least 48h after collection but these data clearly show that the difference in RNA level changes.

Additional comments

The manuscript entitled, Stability of hepatitis B virus pregenomic RNA in plasma specimens under various temperatures and storage conditions” describes how the pgRNA level differs regarding different storage conditions. This subject is important as pgRNA reflects viral replication activity and is valuable for monitoring the effect in patients receiving novel anti-HBV therapies. Nevertheless, different storage conditions were supposed to affect RNA levels and in consequence, interfere with the quantification results. This study demonstrated that HBV pgRNA is quite stable under different storage temperatures.

Minor comments:
1) These findings are important but the authors should discuss more significant differences presented in Table 2. The authors claim that pgRNA is stable at room temperature at least 48h after collection but these data clearly show that the difference in RNA level changes.
2) How was the ddPCR lower detection limit calculated?
3) Table 1: what are A and B?

---

## Round 0.2 · accepted · Accept

Thank you for revising your manuscript. You have addressed the concerns adequately.